# PbI$_2$ Single Crystal Growth and Its Optical Property Study

**Der-Yuh Lin [1,\*], Bo-Cheng Guo [1], Zih-You Dai [1], Chia-Feng Lin [2]**  **and Hung-Pin Hsu [3]**

[1] Department of Electronics Engineering, National Changhua University of Education, Changhua 50007, Taiwan; m0653006@gm.ncue.edu.tw (B.-C.G.); konald78440@gmail.com (Z.-Y.D.)

[2] Department of Materials Science and Engineering, National Chung Hsing University, Taichung 40227, Taiwan; cflin@dragon.nchu.edu.tw

[3] Department of Electronic Engineering, Ming Chi University of Technology, New Taipei City 24301, Taiwan; hphsu@mail.mcut.edu.tw

\* Correspondence: dylin@cc.ncue.edu.tw

**Abstract:** In this work, we used the chemical vapor transport (CVT) method to grow PbI$_2$ crystals using iodine as a self-transporting agent. The crystals' structure, composition, and uniformity were confirmed by X-ray diffraction (XRD) and electron probe microanalysis (EPMA) measurements. We investigated the band gap energy using absorption spectroscopy measurements. Furthermore, we explored the temperature dependence of the band gap energy, which shifts from 2.346 eV at 300 K to 2.487 eV at 20 K, and extracted the temperature coefficients. A prototype photodetector with a lateral metal–semiconductor–metal (MSM) configuration was fabricated to evaluate its photoelectric properties using a photoconductivity spectrum (PC) and persistent photoconductivity (PPC) experiments. The resonance-like PC peak indicates the excitonic transition in absorption. The photoresponse I$_{Light}$/I$_{Dark}$-1 is up to 200%.

**Keywords:** PbI$_2$; 2D semiconductor; XRD; PC; PL; Absorption spectroscopy

## 1. Introduction

In recent years, two-dimensional semiconductors (2D), such as graphene, have attracted extensive research attention due to their remarkable characteristics, such as mechanical flexibility and high mobility [1]. The outstanding flexibility of atomic thin (or few layers) 2D materials makes it possible to integrate electronic or photonic devices on unusual substrates, such as plastic and rubber. Zero band gap could be the biggest disadvantage of graphene and limits its application in optoelectronics [2]. Beyond graphene, abundant transition-metal dichalcogenides (TMDCs), such as MoS$_2$ and WS$_2$, the III-IVA group (GaSe and InSe), and the IV-VI group (SnS$_2$ and SnSe$_2$), have adjustable band gaps to overcome graphene's drawbacks and are used in many devices, such as field effect transistors (FETs), photodetectors (PDs) [3], solar cells [4], and sensors [5]. Optoelectronic applications cover a wide range of wavelengths from near infrared to visible light [6].

PbI$_2$ has been extensively studied as low-noise X- and gamma ray detectors at room temperature [7]. Recently, it has gained a lot of attention due to the development of lead halide perovskites, which are promising materials for highly efficient solar cells, light-emitting diodes, and photodetectors [8]. Traditionally, the growth of PbI$_2$ single crystals can be carried out using the Bridgman–Stockbarger method [9] and the Czochralski method [10]. However, some physical and chemical methods, such as physical vapor deposition (PVD) [11], liquid solution method [12], vapor−liquid−solid method [13], and hydrothermal synthesis [14] have been successfully used to synthesize PbI$_2$ nanobelts or nanowires. To date, growing PbI$_2$ single crystals with high chemical and structural quality has been challenging but

essential for high performance detectors. The chemical vapor transport (CVT) method is a traditional and useful method to grow free-standing TMDC $MX_2$ (where M is a transition metal and X is a chalcogen element) bulk crystals [15]. The application of CVT has been extended from growing 3D bulk crystals to synthesizing 2D atomic layers by tuning the growth kinetics. In most cases, the iodide is used as the transport agent to carry transition metal atoms from high temperature zones to low temperature zones.

In this work, the free standing $PbI_2$ crystals were grown without any chalcogen elements. Iodide was used as both a transport agent and reaction element. We have grown the free-standing $PbI_2$ crystals with obvious layered structures using the CVT method and checked the composition using a variety of measurements. The band gap energy was measured using different methods, but the values were not consistent, varying from 2.2 eV to 2.6 eV. We performed a comprehensive study to confirm the band gap energies of bound exciton and free exciton. We also verified the a-lattice and c-lattice constant by X-ray diffraction. Furthermore, we carefully studied the temperature dependence of band gap energies between 20 and 300 K. Moreover, we extracted the temperature coefficients by Varshni equation and O'Donnell and Chen's equation to determine the average phonon energy and lattice thermal conductivity. A simple metal−semiconductor−metal photodetector was used to understand the photoresponse and rise and fall time.

## 2. Materials and Methods

The first step for the growth of $PbI_2$ crystals was to weigh the high purity elements, lead (Pb, 99.99%, Alfa Aesar, Commonwealth of Massachusetts, USA) and iodide (I, 99.99%, Alfa Aesar, Commonwealth of Massachusetts, USA), using an electronic scale to fit the molar ratio of Pb:I to 1:2. The elements were then put into a quartz ampoule (TOCHANCE, Taoyuan City, Taiwan) and sealed in high vacuum ($2 \times 10^{-5}$ Torr) conditions. To obtain an optimal diffusion gradient and suitable temperature for crystal growth, temperatures were set to 750 and 650 °C, respectively, for the two ends of the quartz tube, with a temperature gradient of around 3 °C/cm. After a growth time of 500 h, the temperature was slowly cooled down. During this growth, iodide played the role of a self-transporting agent, carrying the lead atoms from the high temperature zone and performing stable bonding in the low temperature zone. The quality of $PbI_2$ grown was confirmed by XRD, SEM, and EDS.

The crystal structure and lattice constant were examined using an X-ray diffractometer (PANalytical X'Pert Pro,PANalytical, Almelo, Holen, Netherlands) with Cu K$\alpha$ radiation ($\lambda$ = 1.5406 Å). Photoluminescence (PL) measurements were determined using Nanometer Scale Raman PL Microspectrometer (Nanofinder 30, Tokyo Instruments, Tokyo, Japan) with the laser operating in the wavelength of 488 nm. The reflection measurement was taken using the microspectrometer (OtO handheld microspectrometer, OTO Photonics, Hsinchu City, Taiwan). For the temperature-dependent absorption measurements, a 130 W tungsten-halogen lamp equipped with a monochromator was used to provide monochromatic light. The samples were mounted on a closed-cycle cryogenic refrigerator (Janis Research CCS-250, Janis Research, Woburn, MA, USA) equipped with a digital thermometer controller (Cryocon Model 32B, Cryogenic Control Systems, Rancho Santa Fe, CA, USA) for measuring the absorption spectra in the temperature range of 20 to 300 K. A mechanical chopper (Model SR540 Chopper controller, Stanford Research System, California, USA) was used to switch the monochromatic light between ON and OFF states in a frequency of 10 Hz. A silicon detector (UDT-555D, OSI Optoelectronics, California, USA) was placed behind the sample to detect the transmitted light intensity and transfer it to a voltage signal, feeding it to a lock-in amplifier (AMETEK Scientific Instruments 7265, AMETEK, Commonwealth of Pennsylvania, USA). For the photoconductivity (PC) spectrum measurement, a sourcemeter (Keithley 2400, Keithley, Cleveland, OH, USA) was used to provide a constant voltage (20 V) across the sample. The photoexcited current was recorded by a lock-in amplifier through a current to voltage transfer circuit.

## 3. Results

*Subsection*

Figure 1a,b shows the SEM and EDS images of PbI$_2$ crystal. The red square area has been enlarged and presented in the right figure. In this enlarged SEM image, we can clearly find the layered structure with 120° corner, which indicates that the PbI$_2$ crystals exhibit hexagonal crystallites [16,17]. Each PbI$_2$ layer exhibits I–Pb–I repeating units and stacks along the c-axis with van der Waals interactions between neighboring layers. The left SEM image reveals that the PbI$_2$ crystal has a flat surface on a layer and the EDS images confirm that Pb (purple dots) and I (green dots) elements are uniformly distributed in the crystals. The ratio of Pb and I elements given as atomic percentage is determined to be 36:64. This value has also been confirmed by EPMA, with the result given as 32.3:67.7. These results confirm the PbI$_2$ crystals with a molar ratio very close to the ideal value 1:2.

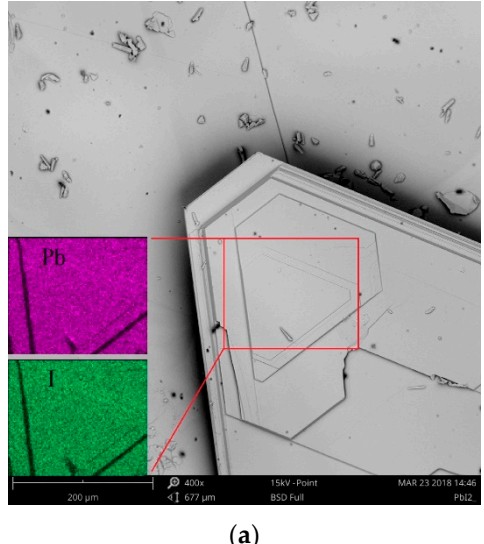

(a)

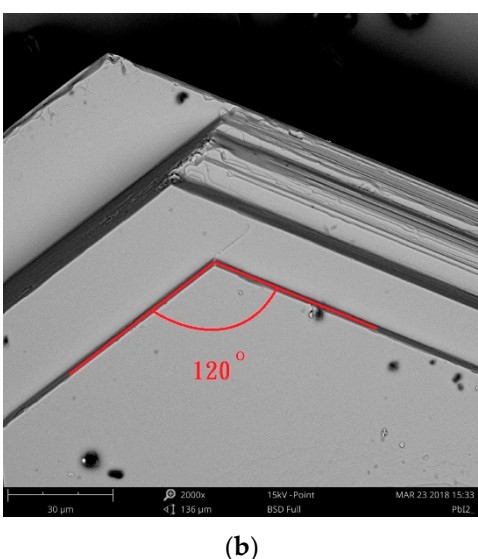

(b)

**Figure 1.** (**a**) SEM and EDS images (inserts) of PbI$_2$ crystal. (**b**) SEM image of the red square area in the left figure. This SEM image clearly shows the layered structure and corner angles of hexagonal crystals.

The structure of PbI$_2$ crystal was characterized by the XRD measured in 10 to 60 degree angle range. The XRD result is shown in Figure 2; the diffraction peaks observed at angles (2θ) 12.70, 25.41, 38.67, and 52.35 degrees correspond to planes (001), (002), (003), and (004) from left to right, respectively. Because all of the peaks are derived from the same planes of symmetry (001), which are perpendicular to the c-axis, this result confirms the single crystalline nature of our PbI$_2$ crystal [18]. Based on the Bragg equation λ = 2dsinθ and using the λ of Cu Kα (λ = 1.5406 Å), the value of the (001) peak d means the lattice constant of c-axis can be calculated to be 6.964 Å. Our experimental c-lattice constant matches well with the published data c_2H = 6.9862 Å for PbI$_2$ and also confirms the PbI$_2$ is stacked in a 2H structure [19]. The XRD pattern of PbI$_2$ powder is shown in Figure 2; the 2-Theta degrees have been transformed to the corresponding degrees of λ = 1.5406 Å using $2\theta_2 = 2sin^{-1}\left(\frac{\lambda_2}{\lambda_1}\sin(\theta_1)\right)$ for easy comparison. The result demonstrates that all the peaks were indexed to the pure hexagonal phase PbI$_2$ structure (JCPDS card no.79-0803), with the space group of P-3m1 (164), and the lattice constant of a is calculated to be 4.56 Å, which is in agreement with previous work [20].

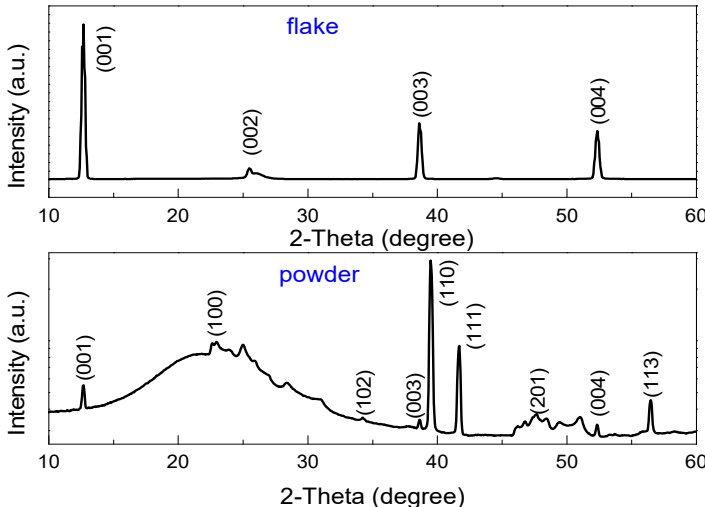

**Figure 2.** XRD patterns of PbI$_2$ crystal characterized and powder in the angle range between 10 and 60 degrees.

In order to identify the band gap energy, we performed the transmittance and photoluminescence (PL) measurements at 300 K. At sub-band gap photon energies, a semiconductor is basically transparent, meaning that no photon absorption happens in this energy range and most of the incident light can pass through the sample. A semiconductor will gradually become opaque when the photon energy closes to the band edge, and most of the incident light will decay in the sample. The Beer–Lambert law relates the attenuation of light to the absorption of the material through the equation $I(x) = I_0 e^{-\alpha x}$, where $I_0$ is intensity of incoming light, $\alpha$ is absorption coefficient, and x is distance to the surface. The absorption data in Figure 3 are derived from transmittance experiments through the Beer–Lambert equation, $\alpha d = \ln(I_0/I)$. For a direct transition semiconductor, the Tauc plot of absorption coefficient was plotted in their squared values versus photon energy as shown by the black curve in Figure 3. The band gap energy can be determined by extrapolating the linear region to zero absorption in the lower photon energy region and is found to be 2.349 eV, as shown by the red dotted line. In Figure 3, we also present the PL spectrum, which is a useful method to identify the band gap energy of a direct semiconductor. The inset shows the PL spectrum and two Gaussian fitting curves with peak positions at 2.35 and 2.43 eV, respectively. We found that the low energy PL peak matches well with the absorption band gap energy. We believe it can be attributed to the bound excitonic emission. This assumption was confirmed by PC measurement and is discussed later. The high energy emission peak at 2.43 eV refers to the free excitonic recombination. On the basis of PL and absorption measurements, we demonstrate that our PbI$_2$ crystal is a direct semiconductor [21].

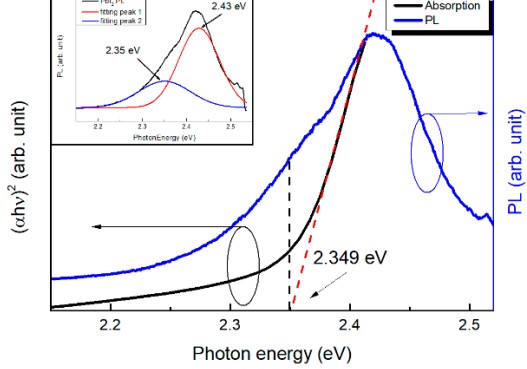

**Figure 3.** Absorption spectra and photoluminescence (PL) spectra of PbI$_2$ single crystal. The inset shows the PL spectra with two fitting curves.

For the purpose of studying the temperature dependence of band gap energies, we performed the temperature-dependent transmittance measurements in the temperature range between 20 and 300 K. Figure 4a shows the absorption spectra measured at different temperatures, and the inset figure shows the comparison of reflectance and absorption spectra. When incident light strikes the sample surface, it may be reflected, transmitted, or absorbed, depending on the photon energy. The relative amounts of reflection and transmission are determined by the refractive indices of the two media and the angle of incidence. Here, we can find that the reflectance curve goes downward as the absorption curve goes upward around the band gap energy. These absorption spectra show a blue shift in photon energy when the temperature decreases from 300 to 20 K. The band gap energies can be extracted by extrapolating the linear portion of these curves to zero. The values corresponding to direct band gaps as a function of temperature is presented in Figure 4b.

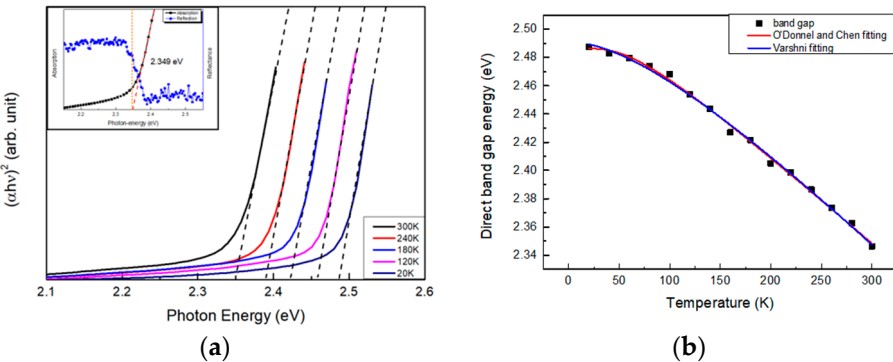

(a)                                                       (b)

**Figure 4.** (**a**) Temperature-dependent absorption spectra measured in the temperature range between 20 and 300 K. (**b**) Band gap energy of $PbI_2$ crystal as a function of temperature. Solid squares represent the experimental values deduced at different temperatures ranging from 20 to 300 K. Solid blue line and dotted red line are the fitting according to Equations (1) and (2), respectively.

The relationship between temperature and band gap energy is usually expressed by the Varshni empirical relation, which is given by:

$$E_g(T) = "\ E_g(0) - (\alpha T^{2}")/(\beta + T),\qquad(1)$$

where $E_g(0)$ is the band gap energy at 0 K, and $\alpha$ and $\beta$ are characteristic parameters of a given material. The constant $\alpha$ is related to the electron–phonon pair, and $\beta$ is related to the Debye temperature. The fitted result is indicated by the solid blue line in Figure 4b. Here, we find that $E_g(0)$ is 2.491 eV and $\alpha$ and $\beta$ are $7.35 \times 10^{-4}$ eV/K and 161 K, respectively. The fitted values of $E_g(0)$, $\alpha$, and $\beta$ are listed in Table 1. In Table 1, we find the parameters $\alpha$ and $\beta$ match well with the data in [22].

**Table 1.** Values of the band gap energy at 0 K, and the characteristic parameters $\alpha$ and $\beta$ according to Equation (1).

|        | $E_g(0)$ (eV) | $\alpha$ (meV/K) | B (K) | Ref. |
|--------|---------------|------------------|-------|------|
| $PbI_2$ | 2.491 | 0.7349 | 161.4 | This work |
| $PbI_2$ | 2.49 | 0.705 | 130 | [22] |
| GaP | 2.338 | 0.62 | 460 | [23] |
| $SnS_2$ | 2.56 | 2.12 | 797 | [24] |

Compared to other materials, such as GaP and $SnS_2$, $PbI_2$ has a low $\beta$ value, implying it has a low Debye temperature [25]. In general, Debye temperature has a linear correlation with lattice thermal conductivity [26]. This result shows $PbI_2$ has the lowest lattice thermal conductivity, which enables it to have a high thermoelectric figure of merit, giving it great potential in thermoelectric applications [27,28].

Another useful numerical least-square fitting equation proposed by O'Donnell and Chen [29] is given by:

$$Eg(T) = Eg(0) - S\langle\hbar\omega\rangle[\coth(\langle\hbar\omega\rangle/(2k_BT)) - 1], \tag{2}$$

where $E_g(0)$ is the band gap energy at 0 K, S is a dimensionless coupling constant, $\langle\hbar\omega\rangle$ is an average phonon energy, and $k_B$ is the Boltzmann constant. The fitted result is indicated by the solid red line in Figure 4b. For $PbI_2$, we find that $E_g(0)$ is 2.485 eV, S = 3.6, and $\langle\hbar\omega\rangle$ = 13.1 meV. The fitted results are also listed in Table 2.

**Table 2.** Values of the band gap energy at 0 K, the fitted dimensionless coupling constant S, and average phonon energy $\langle\hbar\omega\rangle$, according to Equation (2).

|  | $E_g(0)$ (eV) | S | $\hbar\omega$ (meV) | Ref. |
|---|---|---|---|---|
| **$PbI_2$** | 2.485 | 3.60 | 13.1 | This work |
| **$PbI_2$** | 2.486 | 3.66 | 15.6 | [22] |
| **GaP** | 2.338 | 3.35 | 43.6 | [23,30] |
| **$SnS_2$** | 2.559 | 5.76 | 26 | [24] |

The temperature dependence of band gap energy is one of the fundamental properties of semiconductors. Major contributors to the variation of band gap energy with temperature are the effects of lattice dilation and electron−phonon interaction [31]. In solid crystals, phonons play an important role in many of the physical properties of condensed matter, such as thermal conductivity. It has been demonstrated by theoretical calculations and experimental results that the variation of band gap energy is almost linear with temperature at a high temperature range, which can help to extract the phonon characteristic parameters. In our study, the average phonon energy extracted from fitting to Equation (2) is 13.1 meV, which matches well to the longitudinal–optical (LO) phonon energy ($\omega_{LO}$ = 13.4 meV) of $PbI_2$ [32]. In Table 2, we also find the parameters S and $\langle\hbar\omega\rangle$ match well with the data reported in Ref. 29.

Photoconductivity is a widely used technique for understanding the optical and electrical interaction in semiconductors. In Figure 5, we collected the absorption and PC curves together, measured at 300 K. When the incident photon has energy close to the band gap energy, it could excite an electron transition from valance band to conduction band. As a result, the electrical conductivity increases due to the photoexcited carriers, including electrons and holes. The number of photoexcited carriers increases in proportion to the absorption of incident light and reaches its maximum at the band gap energy. Since the absorption response continues at higher energies, the PC spectrum generally follows a step. In contrast to the PC spectrum with a continued absorption response at the high energy range, it is worth noting that we observed an apparent peak located around the band gap energy, in this study. This peak shape, like an exciton resonance peak, indicates that the excitonic transition from valance band to conduction band may dominate the light absorption process in our $PbI_2$ samples.

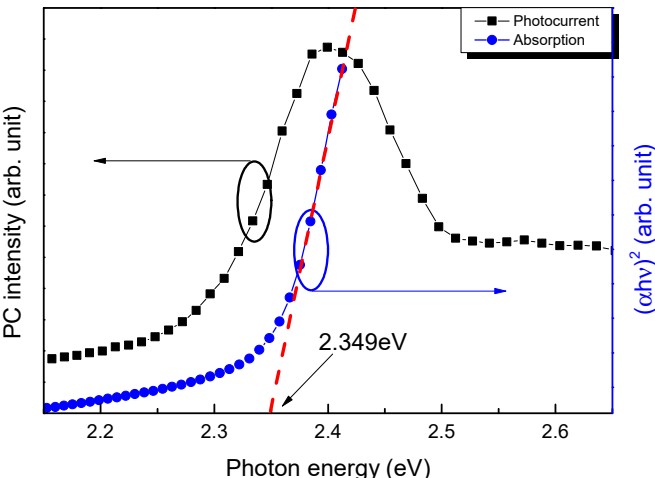

**Figure 5.** Band gap energy of PbI$_2$ crystal as a function of temperature. Solid squares represent the experimental values deduced at different temperatures ranging from 20 to 300 K. The solid blue line and dotted red line are the fitting according to Equations (1) and (2), respectively.

To understand the photoresponse of PbI$_2$, we prepared a metal–semiconductor–metal (MSM) photodetector using silver glue as metal contact to the PbI$_2$ crystal. Figure 6 presents the persistent photoconductivity (PPC) result excited by a solid laser diode (520 nm) operated at a frequency of 500 Hz. The current in dark conditions is about 12 nA under 20 V bias voltage. This current increases to 37.5 nA after illumination and persists for a certain period of time. When the light source is subsequently terminated, the photocurrent exhibits an exponential decay. The rise or fall time of a photodetector is defined as the time for the photocurrent to rise or fall from 10% to 90% or 90% to 10% of the final value, respectively. For a photodetector, it would be good to have a fast response as light illuminates on its surface and it has a short recovery time as the light turns off. The PPC curve shows the rise time is 0.15 ms, and the fall time is 0.11 ms. Furthermore, the photoresponse I$_{Light}$/I$_{Dark}$-1 is up to 200%.

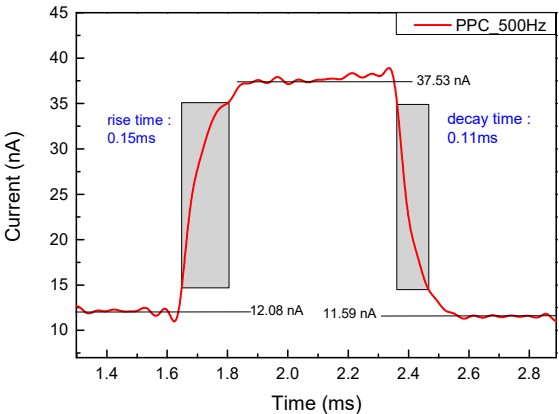

**Figure 6.** Persistent photoconductivity (PPC) curve of PbI$_2$. This time-dependent photoresponse shows the rise time, fall time, and photoresponse.

## 4. Conclusions

We have demonstrated the growth of free standing PbI$_2$ crystals using the CVT method. The PbI$_2$ crystals exhibit hexagonal crystallites and stack along the c-axis in a 2H structure by van der Waals forces between neighboring layers. The band gap energies between 20 and 300 K have been studied by the absorption spectra. The temperature coefficients have been extracted via the Varshni equation and O'Donnell and Chen's equation. We found that PbI$_2$ has very low lattice thermal conductivity and an average phonon energy of 13.1 meV. These properties enable it to have a high thermoelectric

figure of merit. We performed a MSM photodetector and determined its rise time and fall time as 0.15 and 0.11 ms, respectively; the photoresponse is up to 200%. Our results have demonstrated $PbI_2$ is a potential thermoelectric material and also a suitable material for photodetector in visible range.

**Author Contributions:** Conceptualization, D.-Y.L. and B.-C.G.; Data curation, B.-C.G. and Z.-Y.D.; Formal analysis, D.-Y.L. and B.-C.G.; Funding acquisition, D.-Y.L.; Investigation, C.-F.L. and H.-P.H.; Methodology, C.-F.L. and H.-P.H.; Supervision, D.-Y.L.; Resources, C.-F.L. and H.-P.H.; Writing—original draft preparation, B.-C.G.; Writing—review and editing, D.-Y.L.

**Funding:** This study was supported by the Ministry of Science and Technology of Taiwan, Republic of China, under contract MOST 107-2112-M-018- 002 and MOST 108-2221-E-018-010.

**Acknowledgments:** We thank Chung-Kai Chang and National Synchrotron Radiation Research Center (NSRRC) for assistance in performing the XRD measurement.

**Conflicts of Interest:** The authors declare no conflict of interest.

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
