# Peer review of "PbI2 Single Crystal Growth and Its Optical Property Study"

_crystals, doi:10.3390/cryst9110589_

Round 1

Reviewer 1 Report

Der-Yuh Lin and colleagues grew PbI2 single crystals by Chemical Vapor Transport method and characterized them. The work is interesting and is recommended for publication after addressing a few concerns:

Can the authors perform photoluminescence (PL) studies on the crystals? PL is a more direct method of measuring the bandgap and may indicate, in comparison with absorption spectra, if the PbI2 is a direct semiconductor or indirect. The introduction is very broad and is a bit irrelevant. It is unclear what was known about PbI2, what was missing and what the authors added. The authors are recommended to rewrite the Intro in order to highlight the missing gaps and novelty.

Author Response

Reviewer1’s comments

Der-Yuh Lin and colleagues grew PbI2 single crystals by Chemical Vapor Transport method and characterized them. The work is interesting and is recommended for publication after addressing a few concerns:

Can the authors perform photoluminescence (PL) studies on the crystals? PL is a more direct method of measuring the bandgap and may indicate, in comparison with absorption spectra, if the PbI2 is a direct semiconductor or indirect. The introduction is very broad and is a bit irrelevant. It is unclear what was known about PbI2, what was missing and what the authors added. The authors are recommended to rewrite the Intro in order to highlight the missing gaps and novelty. 

Reply to the reviewer1’s comments

Thank you very much for your comments. We have performed the PL study and add the result into this manuscript in figure 3. Compared with the absorption and PC results, we found that PbI2 is a direct semiconductor with a bound exciton absorption edge at 2.35 eV and a free exciton band gap at 2.43 eV, respectively. We also rewrite the introduction part to highlight our contributions and novelty. The modified paragraphs are indicated in red color.

Reviewer 2 Report

The manuscript under examination, "PbI2 single crystal growth and its optical property study", by Lin et al. is basically a characterization of PbI2 crystals grown by chemical vapour transport (CVT).

I feel that the manuscript may spark moderate interest at best, as the only significant contribution is the growing technique. Apart from that, it basically lacks novelty, as it is mostly confirms already known results.

I actually enjoyed the didactic parts, for instance where the authors take the time to explain which equation they use and its meaning. Regrettably, I also feel that this is not enough for a manuscript to be accepted for publication on Crystals.

Author Response

Reviewer2’s comments

The manuscript under examination, "PbI2 single crystal growth and its optical property study", by Lin et al. is basically a characterization of PbI2 crystals grown by chemical vapour transport (CVT).

I feel that the manuscript may spark moderate interest at best, as the only significant contribution is the growing technique. Apart from that, it basically lacks novelty, as it is mostly confirms already known results.

I actually enjoyed the didactic parts, for instance where the authors take the time to explain which equation they use and its meaning. Regrettably, I also feel that this is not enough for a manuscript to be accepted for publication on Crystals.

Reply to the reviewer2’s comments

Thank you very much for your comments on the didactic or literature part. Beyond them, in this manuscript, we proposed a self-transport CVT method to grow the PbI2 crystals. The resulting crystals were characterized by using XRD, EPMA, SEM, EDS, PC, PL, reflection spectroscopy and variable temperature absorption spectroscopy measurements. From the comprehensive study, we found that PbI2 is a direct semiconductor with a bound exciton absorption edge at 2.35 eV and a free exciton band gap at 2.43 eV, respectively. We have determined the temperature coefficients by the Varshni equation and O’Donnell and Chen’s equation and pointed out PbI2 has very low lattice thermal conductivity and an average phonon energy of 13.1 meV, potentially giving rise to a high thermoelectric figure of merit. We also performed a simple MSM assembly and characterized by photoconductivity spectrum (PC) and persistent photoconductivity (PPC) measurements to demonstrate that PbI2 is not only a potential thermoelectric material but also a suitable material for photodetector in visible range. We add some experimental results and also rewrite the introduction part to highlight our contributions and novelty. The modified paragraphs are indicated in red color. Therefore, we hope you could kindly support this manuscript to be published in Crystals.

Reviewer 3 Report

The manuscript reports the synthesis of high-quality single crystals of PbI2 with the chemical vapor transport (CVT) method. The resulting crystals were characterized by using PXRD, electron probe micro- analyses (EPMA) and variable temperature absorption spectroscopy measurements. The authors have determined the temperature coefficients by the Varshni equation and O’Donnell and Chen’s equation. Apparently PbI2 has very low lattice thermal conductivity and an average phonon energy of 13.1 meV, potentially giving rise to a high thermoelectric figure of merit. The corresponding material was utilized for the assembly of a photodetector. The resulting assembly was characterized by photoconductivity spectrum (PC) and persistent photoconductivity (PPC) measurements. The resonance like PC peak indicates the excitonic transition in absorption. The photoresponse ILight/IDark-1 is up to 200%. The above results indicate that PbI2 is a potential thermoelectric material and also a suitable material for photodetector in visible range. The study is interesting and thorough; therefore, I recommend this manuscript to be published in Crystals after minor revisions.

Additional comments to the authors:

Figure 2, please add the calculated PbI2 pxrd pattern for comparison. CVT method, please add some more details about the applicability of the method. The first paragraph of introduction needs references. Figure 3, please include the recorded absorption or reflectance spectra.

Author Response

Reviewer3’s comments

The manuscript reports the synthesis of high-quality single crystals of PbI2 with the chemical vapor transport (CVT) method. The resulting crystals were characterized by using PXRD, electron probe micro- analyses (EPMA) and variable temperature absorption spectroscopy measurements. The authors have determined the temperature coefficients by the Varshni equation and O’Donnell and Chen’s equation. Apparently PbI2 has very low lattice thermal conductivity and an average phonon energy of 13.1 meV, potentially giving rise to a high thermoelectric figure of merit. The corresponding material was utilized for the assembly of a photodetector. The resulting assembly was characterized by photoconductivity spectrum (PC) and persistent photoconductivity (PPC) measurements. The resonance like PC peak indicates the excitonic transition in absorption. The photoresponse ILight/IDark-1 is up to 200%. The above results indicate that PbI2 is a potential thermoelectric material and also a suitable material for photodetector in visible range. The study is interesting and thorough; therefore, I recommend this manuscript to be published in Crystals after minor revisions.

Additional comments to the authors:

Figure 2, please add the calculated PbI2 pxrd pattern for comparison. CVT method, please add some more details about the applicability of the method. The first paragraph of introduction needs references. Figure 3, please include the recorded absorption or reflectance spectra.

Reply to the reviewer3’s coomments

Thank you very much for your comments and recommendation. We have added the calculation about the lattice constant of a-axis and c-axis which were calculated to be 0.456 nm and 0.68984 nm from pxrd. We also added a paragraph to address the applicability of CVT method. We performed the reflectance spectra and put it in the inset of figure 4(a). We found that the reflectance curve goes downward as the absorption curve goes upward around the band gap energy. We also rewrite the introduction part to highlight our contributions and novelty. The modified paragraphs are indicated in red color.

Round 2

Reviewer 2 Report

After Authors' modifications, the manuscript make it suitable for pubblicationon Crystals